# A possible thermodynamic definition and equation of state for a model of political election cycles

George-Rafael Domenikos[1], Alexander V. Mantzaris[2]*

**1** Centre of Mathematical Sciences, Department of Applied Mathematics and Theoretical Physics, University of Cambridge, Cambridge, United Kingdom, **2** Department of Statistics and Data Science, University of Central Florida (UCF), Orlando, FL, United States of America

☯ These authors contributed equally to this work.
* Alexander.Mantzaris@ucf.edu

## Abstract

This work demonstrates how a simulation of political discourse can be formulated using variables of the agents' behaviors in a simulation, as thermodynamic variables. With these relations the methodology provides an approach to create a correspondence between the variables of an agent based social system and those of a thermodynamic system. Extended from this observation, diagrams akin to a P-V diagram for gases can be created for this social system. The basic thermodynamic variables of temperature, pressure and volume are defined from a system of agents with political and non-political actions engaged in simulated political discourse. An equation of state is defined for the simulated political phenomenon. Through this equation of state the full thermodynamic map of the system is presented under a P-V diagram with isothermal and isentropic lines, which is able to represent the political situation of the system at each point of time. The classic election cycle that takes place can be represented on this thermodynamic map (corresponding to an Otto cycle). This provides a possibility for researching macroscopic social cycles as a thermodynamic/informational cycle as the traces on the thermodynamic map show similarities to an Otto cycle. Such a formulation reinforces the endeavours of social physics to view social phenomena with physical principles.

## Introduction

Political discourse is an essential component of a democratic society which aims to maintain and improve the welfare of its citizens. This involves the citizens themselves taking part in the governance process by casting votes in favor of politicians that support policies they believe to be in their best interest and/or greater good [1–3]. For this process to work correctly there must be a consensus among the voters and this typically involves them sharing their opinions with peers [4]. From a process of healthy political discourse where citizens are seeking a mutual benefit a situation of political polarization can ensue which can grow into catastrophic scenarios [5] motivating the understanding of these systems. There are many useful models

**Funding:** This work was partially supported by the Defense Advanced Research Projects Agency

(DARPA) under agreement HR00112290104 (PA-21-04-06). There was no additional external funding received for this study.

**Competing interests:** The authors have declared that no competing interests exist.

developed recently and each has their merits and insights to offer [6–9]. The work presented here presents a methodological approach on analyzing a simulation of political discourse as a thermodynamic system. It will be seen how the political system variables can correspond to thermodynamic variables which can then define the system dynamics necessary to view elections cycles as transitions of a heat engine.

Prior work of the authors has shown how the entropy trace of a model of residential agents seeking homogeneity (Schelling model) can be calculated along the simulation iterations; [10]. A following article, [11], shows how incorporating a monetary variable into the Schelling model as a dual dynamic allows agents to change position while inducing a spending change resulting in a model that overall increases its entropy value with simulation iterations. Subsequent work in [12], proposes a model where the agents remain in the same spatial configuration but influence the agents in their immediate locality for their support of a political affiliation [13]. In [14] the authors present an approach to take the model of political discourse introduced in [12], and show that it is possible to monitor all the entropic components of the system state whereby showing that the system abides to the 2nd law of thermodynamics. The works of [15, 16] presents a general approach to investigating the thermodynamic quantities of that states within "Conway's Game of Life" in which the temperature of the system is calculated.

In the classic Schelling model each agent resides within a cell on a grid, is allocated a random political affiliation and their locality is their immediate surrounding cells [17, 18]. The Ising model of ferromagnetism [19]) utilizes a similar locality metric and the overlap between the two models is a source of inspiration for social physics [20, 21]. The political affiliation utilized here is modeled as a bipartisan spectrum [22] where each side is designated as values residing in the positive or negative domain with zero being neutral. At each simulation iteration the agents update their affiliation with the aggregate values from their neighbors with their own value. Using the probability distribution for the macrostates of the political affiliations across the agents the entropy values can be found at each step (introduced in [23]). A key aspect of the model is that when an agent is surrounded by other agents in agreement with its own political affiliation it is considered to be no longer engaged in the activity of political discourse increasing the activity number of non-political actions (peripheral).

In simulations of the system the interchange between peripheral activities and political engagements for a constant total number of actions can be produced. This diagram produces a set of parallel lines for the different number of total activities agents can take part in. Such a diagram is reminiscent of a P-V diagram for thermodynamics where contours separate isothermal lines, and this analogy aligns within the objective of *Social Physics* [24–29]. As will be shown the average of the total number of activities among the agents will correspond to the temperature of the system [30]. The peripheral activities will correspond to the pressure, and the political engagements to the volume. This provides the ability to create a full thermodynamic map through these fundamental variables. Based on the behaviors and the interconnections of the entropy and the temperature, pressure and volume, the equation of state of the system describing this political simulation is produced. As a result the set of trajectories can cover areas on a thermodynamic map as is done by thermal engines. The thermodynamic cycle is akin to an Otto cycle as will be demonstrated and discussed in the Results section. A key aspect from this research which differs from previous work is that it allows the basic macroscopic variables of an agent based system (social system) to be considered within a thermodynamic context. From this the subsequent aspects of the model can then be viewed as changes on those basic quantities and therefore monitored as a thermodynamic system producing similarities to other thermodynamic processes such as the Otto cycle which will be displayed. This differs from previous research which did not consider the thermodynamic

quantities in isolation to produce a model which can then be used to map each macrostate to a position in the thermodynamic map, or which would allow isothermal and isentropic lines to be produced.

## Methodology

The model used in this paper is based on a previously proposed model in [14] which is a neighbor interaction model of political discourse. In this model a $N \times N$ lattice is used to describe the locations of the agents (one in each cell), with no empty cells. Each of the agents has a political affiliation value at any time point. The definition assumes a bipartisan ideological framework, and each of the possible votes is described as a 0 or 1 at the cells of the agents. A matrix is formed from the voting affiliation values, $C$, for each agent at their lattice positions for each time point. The matrix, $M$, contains the value for the voting action of each agent in the lattice at each time point. Another matrix which holds the state of the ideological mismatch of an agent with their locality is produced, $I$. This information is used to find the political actions of the agents and the definitions follow that at each timestep $t \in [1, \ldots, T]$ the values of the $C$ matrix are calculated using Eq 1 for every position $i, j$ in the lattice using the $t - 1$ values. Based on these calculations, values for $M$ can be produced according to the definition from Eq 1. In the special case when $C_{i,j,t} = 0$ then the value $M_{i,j,t-1}$ is retained. The simulation presented in this work operates under a $C_{max} = 4$, meaning that the range of the values of the C matrix is $C_{i,j} \in \{-C_{max}, \ldots, C_{max}\}$. The $I$ matrix is used to describe the local ideological inhomogeneity that each agent experiences. By using these three variables all the information about the political state of an agent are defined for each timepoint ($C_{i,j,t}$, $M_{i,j,t}$, $I_{i,j,t}$). The matrix values are defined as:

$$C_{i,j,t} = \sum_{m=i-1}^{i+1} \sum_{n=j-1}^{j+1} C_{i,j,t-1} + \begin{cases} +1 & \text{if} \quad C_{m,n,t-1} > 0 \land C_{i,j,t-1} < C_{max} \land (i \neq m \land j \neq n) \\ -1 & \text{if} \quad C_{m,n,t-1} < 0 \land C_{i,j,t-1} > -C_{max} \land (i \neq m \land j \neq n) \\ 0 & \text{if} \quad C_{m,n,t-1} = 0 \land (i \neq m \land j \neq n) \\ 0 & \text{if} \quad (m < 1) \lor (n < 1) \lor (m > N) \lor (n > N) \end{cases}, \quad (1)$$

$$M_{i,j,t} = \begin{cases} 1 & \text{if} \quad C_{i,j,t} > 0 \\ 0 & \text{if} \quad C_{i,j,t} < 0 \end{cases}, \quad (2)$$

$$I_{i,j,t} = \sum_{m=i-1}^{i+1} \sum_{n=j-1}^{j+1} \begin{cases} +1 & \text{if} \quad M_{m,n,t} = 1 \land (i \neq m \land j \neq n) \\ -1 & \text{if} \quad M_{m,n,t} = 0 \land (i \neq m \land j \neq n) \\ 0 & \text{if} \quad C_{m,n,t} = 0 \land (i \neq m \land j \neq n) \\ 0 & \text{if} \quad (m < 1) \lor (n < 1) \lor (m > N) \lor (n > N) \end{cases}. \quad (3)$$

Independent and initially randomized simulations of the system are run and produce a distribution for the macrostates of the system at each time point. These Monte-Carlo samples are taken to produce the probabilities for the finding agents in a particular state at a given time from which the entropy can then be found.

As a result of the state of an agent's locality (based upon values stored in $C$, $M$, $I$) an agent is deemed to be either active in political discourse or not. In order to find the number of the

political actions per agent the following equation is proposed:

$$
n_{pol_{i,j,t}} = \begin{cases} 1 & \text{if} \quad (I_{i,j,t} \neq 0) \wedge |C_{i,j,t}| \neq C_{max} \\ 0 & \text{if} \quad (I_{i,j,t} C_{i,j,t} < 0) \wedge (|C_{i,j,t}| = C_{max}). \\ 0 & \text{if} \quad I_{i,j,t} = 0 \end{cases}
\tag{4}
$$

This quantity decreases when agents are positioned in ideological localities which agree with their own. It is considered that an agent can engage in a finite number of actions in each time point which can be allocated towards political discourse (political actions, $n_{pol}$) and peripheral activities (non-political actions, $n_{peripheral}$). Through the total number of actions an agent can take per time point, and the state of the political engagement; the number of peripheral actions (non-political) is found via:

$$
n_{peripheral_{i,j,t}} = n_{total_{i,j,t}} - n_{pol_{i,j,t}}.
\tag{5}
$$

Here $n_{total_{i,j,t}}$ is distributed according to a truncated binomial distribution, $Binom(n_{total}; n_{max}, 0.5)$ where $n_{max} = 7$ and this is the maximum number of actions an agent is assumed to perform per time point.

The equation used to find the probability for whether an agent is engaged in political discourse, or not, at a time point $t$ is found via:

$$
p_{pol}(v_{pol}, t) = \frac{1}{N_{sim}} \sum_{k=1}^{N_{sim}} \left( \frac{1}{N^2} \sum_{i=1}^{N} \sum_{j=1}^{N} \delta_{v_{pol}, n_{pol_{i,j,t,k}}} \right),
\tag{6}
$$

where $v_{pol} \in [0, 1]$ (non-active or active). This is the expected value for agent political activity across all agents in the lattice. It is then possible to find the entropy for the political action:

$$
S_{v_{pol}, t} = -\sum_{v_{pol}} p_{pol}(v_{pol}, t) \ln(p_{pol}(v_{pol}, t)).
\tag{7}
$$

This approach follows the standard definition of the Shannon entropy [31–33].

The maximum entropy arises when the least knowledge for the actions of the agents occurs. To calculate this theoretical possibility it is assumed that every action is different and unknown, and thus the probability for each action to take place is:

$$
p_{action} = \frac{1}{n_{total}}.
\tag{8}
$$

Here $n_{total}$ signifies the maximum number of actions an agent can partake in per point of time and is calculated at each time point independently. At each simulation time point $t$, the entropy for the total actions of the system can be calculated using:

$$
S_{total} = -\sum_{1}^{n_{total}} p_{action} \ln(p_{action}).
\tag{9}
$$

Deriving the entropy for the peripheral actions can then be computed from:

$$
S_{peripheral,t} = S_{total,t} - S_{political,t}.
\tag{10}
$$

The $S_{total}$ is the sum of the two entropies, $S_{peripheral}$ and $S_{political}$, and it has a specific value given the result of the distribution at every simulation.

## Thermodynamic parallel

As a result of the formulation for political discourse the second law of thermodynamics [34] is respected since the entropy values for the overall system ($S_{total,t}$) remains constant. Consequentially, the first law of thermodynamics [35] is also then preserved. Having a model that respects these laws of thermodynamics allows for further inspiration to be drawn from other basic physical models in how the system can be perceived in a social physics setting.

This behavior of the system can produce an analogue to the thermodynamic behavior of a gas. A gas at a given temperature can be in different states depending on the pressure and the volume. For any gas a P-V diagram can be produced where one can observe the isothermal lines [36, 37], showing how for varying volumes and pressures the temperature remains the same. In this model one can think of the $n_{total}$ as the variable corresponding to the temperature (amount of activity an agent/citizen conducts per time unit). This is similar to that of a gas where the temperature, using the distribution function, defines the energy states that the particles of the gas can occupy. The greater the pressure in the gas the lesser the volume has to be for it to remain at a steady temperature (inverse proportionality). Analogously in this model, for a given number of actions per time unit, the system can change the allocation between political engagements and peripheral activities also with an inverse proportionality. In this way the political actions of the agents can be thought of as the pressure of a gas and the non-political actions as the volume.

From this analogy the contour lines for a system with a particular value of $n_{total}$ can be drawn with the political and non-political actions on the x & y axes as is commonly done with P-V diagrams. In such a graph the lines for each $n_{total}$ can be thought of as being 'isothermal' where the exchange between the values along the x-y axis occur. Fig 1 shows the result of viewing the system from this perspective. Each line is produced using a different $n_{total}$ (different

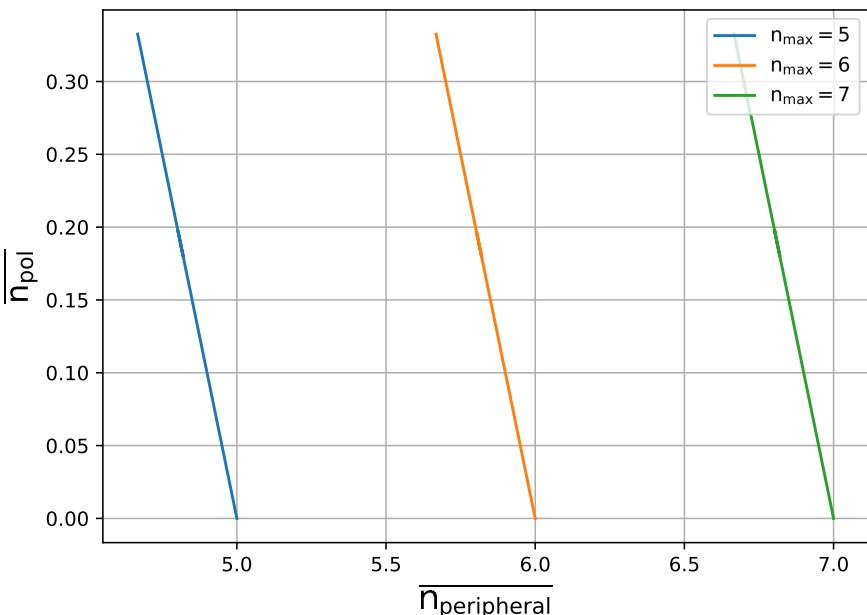

**Fig 1. Isothermal lines for sociological variables.** The lines represent different values of $n_{total}$ which are the total number of actions an agent can perform per unit time. The horizontal axis is the average number of non-politically aligned actions $n_{peripheral}$ and the vertical the average number political actions taken by agents across the grid (political engagement). This relationship between the variables is interpreted as analogous to the isothermal lines in a thermodynamic volume to pressure diagram.

temperatures as being more or less active agents), the horizontal axis is the average number of non-politically aligned actions for the agents (peripheral actions), and the vertical axis the average number of political actions (political engagements $n_E$) per agent in the system.

## Definition of temperature, pressure, volume and equation of state

In this system the temperature needs to be associated with the average number of actions that an agent can partake in, $\widetilde{n_{total}}$. The volume and the pressure are two variables that need to be defined such that they are inversely proportional. An initial assumption was made for the Volume to be the number of political actions and the Pressure to be the number of peripheral actions. Thermodynamically this leads to an equation of state of the form $T \sim P + V$ and is explored in Fig 1. Although interesting to see how any sort of thermodynamic analogue can be produced from a social system it is not in the usual form of thermodynamic equations of state [38, 39]. This form cannot lead to a thermodynamic map like the ones typically encountered when examining gases or fluids [40]. To overcome this a product between the Pressure and the Volume in the equation of state is needed. While respecting the dynamics of the system; the temperature, pressure and volume are therefore considered to be the natural exponential of the total actions, the political actions and the peripheral actions respectively. In this manner a product between the pressure and the volume in the equation of state is achieved, while the constant sum of the peripheral and the political actions is equal to the total number of actions. The temperature is defined as

$$T = e^{\widetilde{n_{total}}} \tag{11}$$

where $\widetilde{n_{total}}$ is the expected value of the total number of actions across agents. The pressure is defined as

$$P = e^{\widetilde{n_{peripheral}}} , \tag{12}$$

and the volume via:

$$V = e^{\widetilde{n_{political}}} . \tag{13}$$

The equation of state is then in the form:

$$T = P V. \tag{14}$$

Having the fundamental equation of state (EOS) Eq 14, the overall behavior of the system can be presented with a thermodynamic map. Given the variables used in this study the Pressure to Volume thermodynamic diagram is chosen as the map for the system which corresponds to the relationship of Peripheral to Political actions respectively. Each simulation state produces a $P$ and a $V$ value pair from which the entropy $S$ and temperature $T$ can be found. From the collection of points the isentropic and isothermal lines can be found and visualized (shown in the results section). In the proposed model the variable that is defined by the simulation of the system is the number of political actions. Thus, similarly to how when describing an engine the volume is the defined variable and the pressure is adapting according to the equation of state, here the political actions were chosen to correspond to the volume and the peripheral actions to the pressure in order to stay true to the analogy between this political simulation system and a real thermodynamic system.

Having defined an equation of state for the social system enables the prediction of all the possible combinations of pressure and volume values. Through this definition a thermodynamic map can be created, as it will be showcased in the results section Fig 2. The representation of the social system in such a way gives the ability to represent different social or political

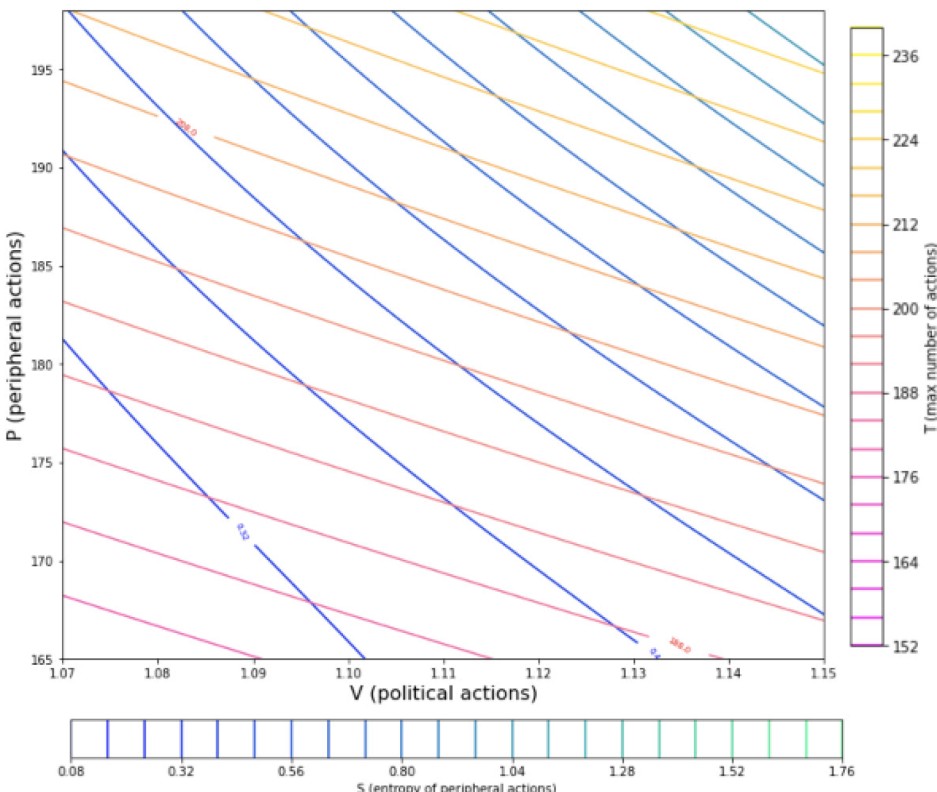

**Fig 2. Thermodynamic map of a social simulation.** Shown are the 'isothermal' and 'isentropic' lines for the social simulation defined. The blue-green lines depict isentropic trajectories, and the pink-yellow lines show the isothermal lines where $\widetilde{n_{total}}$ is kept constant.

procedures that take place as thermodynamic procedures on the P-V map. A procedure in the social system that incorporates a steady number of political action would correspond to an isochoric line on the thermodynamic map. Any sequence of political/societal procedures that is periodic in nature would in turn be represented as a cycle in the thermodynamic map.

As described in the Introduction section this model simulates stages in political discourse which correspond to trajectories in a thermodynamic cycle. This description of the state of the political system though the thermodynamic variables enables the understanding of equivalences between thermodynamic and political processes. As such, political phenomena can be described as a sequence of thermodynamic processes and represented in the map. In this case the political process of a 4-year (or just a 4-phase periodic) election cycle is to be studied. Starting off, the simulation phase where the agents are decreasing their $n_{total}$ while politically engaged due to ideological inhomogeneity corresponds to an election phase analogous to 'isochoric cooling'. In the subsequent phase the agents decrease their political engagements as they arrive at ideological consensus through local interactions. This corresponds to an 'isothermal compression' since the expected number of activities (temperature) remains the same, the peripheral activities increase, and the political engagement decreases. After this phase the agents are in a state of ideological homogeneity (minimum amount of political engagement) which is considered to promote an increase in the total number of total activities ($n_{total}$) which corresponds to a rise in the temperature. This transition is analogous to the isochoric heating. Following the stage of isochoric heating, the agents get closer to the 'election phase' and their

political engagements are activated (0 to 1) due to the increase in ideological heterogeneity. This translates to an 'isothermal expansion' in the thermodynamic map until the election phase is reached again (complete cycle) and the ideological consensus is at its minimum (large $n_{pol}$) before the election phase. The described procedures in the model are seen to take place in a time sequence leading to the formation of a cycle (as they cover an area in the thermodynamic map). With the proposed simulation, given the chosen variables for the political and peripheral actions, leads to the description of a 'social Otto cycle', where the stages are analogous to those of the classical Otto cycle in thermodynamics.

## Results

As described in the Methodology the analogy of the peripheral actions to the pressure and the political actions to the volume the thermodynamic map of the peripheral vs political actions is shown in Fig 2. The isentropic lines are shown in (blue-green) and have a greater negative angle than the isothermal lines shown in (yellow-pink) [40–42]. The application of the above thermodynamic definitions to such a system seems to be leading to an overall behavior that is also consistent to established thermodynamic systems by the form of the contour lines.

Fig 3 shows a trajectory of simulation states on the thermodynamic map, where the states of the model of political discourse produce a thermodynamic cycle. The green connected points

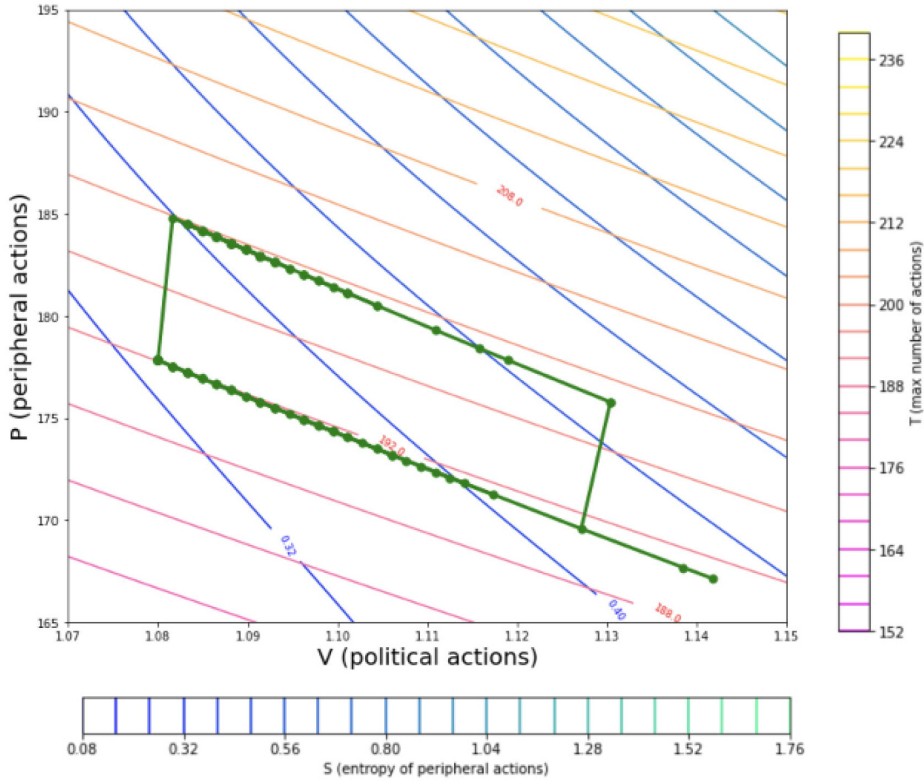

**Fig 3. Ottocycle from points of the steps taken by a model of political discourse.** This plot shows the results of running a set of simulations where the distribution of values of the $\widetilde{n_{total}}$ do not change over time and the political affiliations ($C_{i,j}$) for the agents are changed. The simulation points on the $P - V$ contour map can be seen in green where the isothermal and isentropic lines are displayed. This produces the equivalent of an Otto cycle where the stages are low ideological homogeneity (bottom right), high ideological homogeneity (bottom left), high ideological homogeneity increased activity (top left), low ideological homogeneity (top right), and a return to low ideological homogeneity with lower activity.

display the sequence of simulation states that begin from the bottom right and move counter clockwise. On the bottom right the simulation begins with random values allocated to the agent for their political affiliation values and due to the large heterogeneity in ideological positions in the grid localities there are many political engagements amongst the agents. As the dynamics driving the ideological consensus operate over simulation time, the political actions decrease allowing for the peripheral actions to increase as the number of total actions is constrained to be constant. This produces the lower side of the cycle and is an isothermal trajectory (isothermal compression). The $n_{total}$ constraint value is then increased allowing for more peripheral activities as the political actions remain constant producing the left edge (isochoric heating). In the simulation used to produce this graph a fixed distribution on the $n_{total}$ per agent was used on the two isothermal lines.

For the two isochoric lines the $\widetilde{n_{total}}$ is changed while the average number of political actions ($\widetilde{n_{pol}}$) is kept constant which leads to a higher number of peripheral actions. The upper isothermal line moves in the reverse direction by design. This dynamic represents the trajectory for how a system of agents can gradually move from an expected ideologically homogeneous state to an inhomogeneous state. It should be noted that the isochoric lines may be off of the vertical due to the discrete nature of the simulation.

Fig 4 displays a stochastic simulated trajectory of the model dynamics. The stochastic component is altered so that instead of a fixed sample from the distribution for the number of

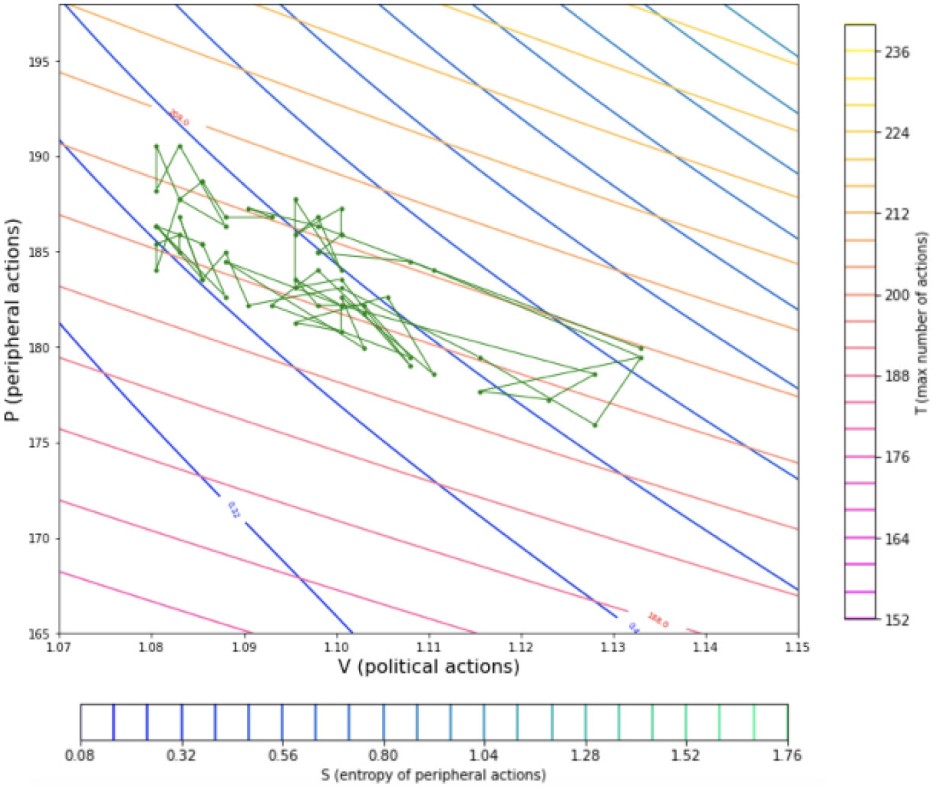

**Fig 4. The stochastic path of the variables of the political system within the thermodynamic map.** The simulation trajectory of the social system with stochasticity in the number of actions per time step, $n_{total}$, is introduced. Shown in green the circular markers denoting states of the simulation. In this figure three full simulation cycles are presented where the maximum number of actions per agent is changed allowing the number of actions to be interpreted as a stochastic variable, temperature, and not as an absolute value shown in Fig 3.

actions of an agent over time this number is sampled at each time step. This aims to represent the randomness of humans in the amount of activity displayed over time individually. Since the number of actions for each agent at each time step is sampled, it is improbable for the $\widetilde{n_{total_t}}$ to be the same and for this reason the isothermal lines deviate from those on the thermodynamic map.

In Fig 5 the black dots depict the number of political and peripheral actions (or volume and pressure) of the timesteps of 8 cycles of the simulation. In applications considering the representations of real Otto cycles of engines the pressure to angle (of the crankshaft) and volume to angle graphs are presented and a function is fitted to the data, in order to then produce the graph of the cycle in the P-V diagram. In the presented case the angle is replaced by the timestep. The simulation, given the nature of the system leading towards homogeneity when left to evolve, offers a lot of data for the average and low political engagements but the outcomes are more scarce considering higher political engagements. This scenario is seen to be lasting only few iterations as seen in [12]. This leads to great difficulty in finding a function that is able to accurately describe the behavior of the system in respect to time as it would be done usually for modelling a real cycle through an experimental engine apparatus [43, 44]. Despite that, observing Fig 4 it is seen that the overall behavior does resemble a cycle and working through the P-V diagram it is possible to create a mathematical representation of the real cycle [45]. To achieve that a cubic spline with 6 control points is fitted to the dots presented in Fig 5, where the squared error is minimised, similarly to applications in [46]. Through this procedure the

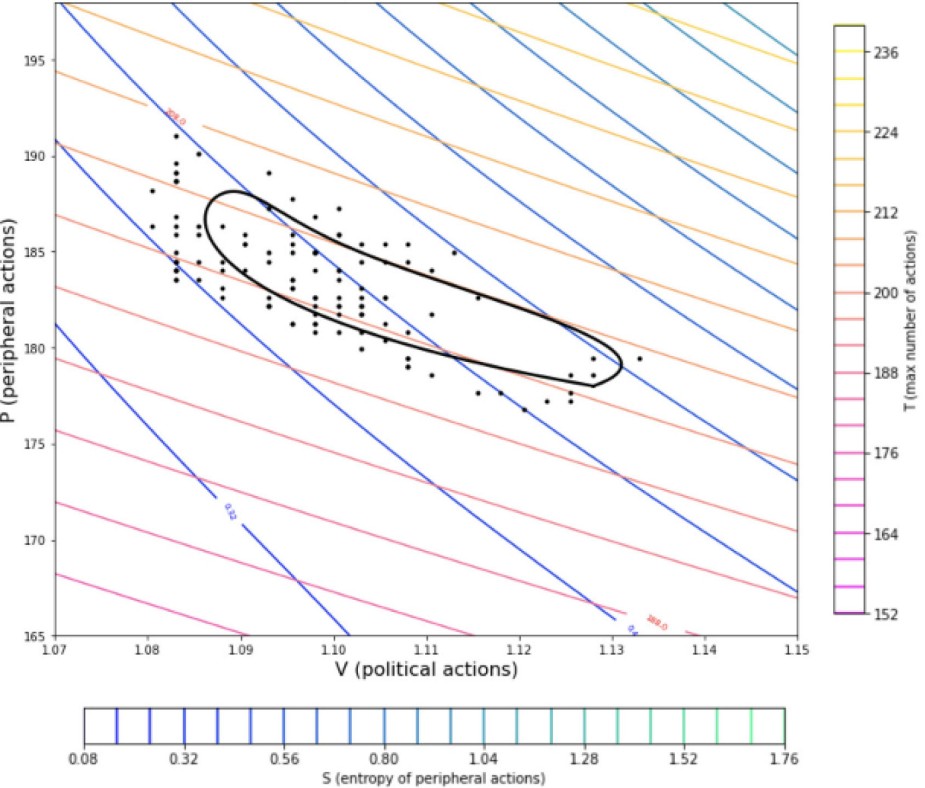

**Fig 5. Mathematical representation of the cycle.** The points in the graph were generated from 6 independently produced stochastic cycles of the system. The approach used here to create this real Otto cycle of the stochastic social system is similar to the method used to produce the real thermodynamic cycles of internal combustion Otto engines through experimental measurements.

black line of Fig 5 is produced, which closely resembles the depictions of real Otto cycles in thermodynamics [47, 48].

## Conclusion

The work presented here covers the exploration of how a simple model of political discourse can be analyzed as a thermodynamic system. From the overall state of the political system it is possible to define variables which correspond to the Temperature, Pressure, and Volume whose values arise from the collective state of the agents. The agents of the system can partake in political discourse or other peripheral activities which are then considered to correspond to Volume and Pressure respectively. The model of political discourse used here has dynamics which drive agents to arrive at ideological consensus via local interactions ([14, 23]). By modelling this political discourse through the thermodynamic variables, it is clear that an equation of state can be made to describe the system. The form of this equation of state (EoS) was carefully chosen as to be in accordance to both the social aspects of the system and also follow the expected thermodynamic behaviors of classical EoS.

By utilizing this formulated equation of state, a full thermodynamic map is created which showcases all the different states that the modeled society can exist in. Having all the possible states of the system represented, the trajectory of the political events/processes can be observed as a trace of points on the thermodynamic map. By utilizing the different isothermal and isentropic lines on the P-V diagram, the processes of a 4-phase political election cycle are described and drawn. The results of this indicate the 4-phase election cycle in this system seems to correspond to a thermodynamic Otto cycle. Future work entails showing how other social cyclic phenomena can be seen as cycles on a thermodynamic map once their EoS is defined. Future work will also entail considering the energy of the system and the correspondence of the temperature with the entropy. Further justification of the usage of thermodynamic cycles methodology in political election cycle shall be attempted with the introduction of the definition of an effective Hamiltonian (that is dependent on $(C, M, I)$), so the Hamiltonian change $dE$ accompanied by the entropy change $dS$ will lead to the temperature dependence expressed by the formula $T = \frac{dE}{dS}$.

## Supporting information

**S1 Data.**
(ZIP)

## Author Contributions

**Conceptualization:** George-Rafael Domenikos, Alexander V. Mantzaris.

**Formal analysis:** George-Rafael Domenikos, Alexander V. Mantzaris.

**Funding acquisition:** Alexander V. Mantzaris.

**Investigation:** George-Rafael Domenikos, Alexander V. Mantzaris.

**Methodology:** George-Rafael Domenikos, Alexander V. Mantzaris.

**Resources:** Alexander V. Mantzaris.

**Software:** George-Rafael Domenikos.

**Supervision:** Alexander V. Mantzaris.

**Validation:** Alexander V. Mantzaris.

**Visualization:** George-Rafael Domenikos, Alexander V. Mantzaris.

**Writing – original draft:** George-Rafael Domenikos, Alexander V. Mantzaris.

**Writing – review & editing:** George-Rafael Domenikos, Alexander V. Mantzaris.

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
