## [Decision Letter · Decision Letter 0]

10 Jan 2024

PONE-D-23-39285A Thermodynamic Definition and Equation of State for a Model of Political Election CyclesPLOS ONE

Dear Dr. Mantzaris,

Thank you for submitting your manuscript to PLOS ONE. After careful consideration, we feel that it has merit but does not fully meet PLOS ONE’s publication criteria as it currently stands. Therefore, we invite you to submit a revised version of the manuscript that addresses the points raised during the review process.

We look forward to receiving your revised manuscript.

Kind regards,

Mohammad Tariq

Academic Editor

PLOS ONE

Journal Requirements:

This work was partially supported by the Defense Advanced Research Projects Agency (DARPA) under agreement HR00112290104

(PA-21-04-06).

4. Please upload a new copy of Figure 1 as the detail is not clear. Please follow the link for more information: https://blogs.plos.org/plos/2019/06/looking-good-tips-for-creating-your-plos-figures-graphics/" https://blogs.plos.org/plos/2019/06/looking-good-tips-for-creating-your-plos-figures-graphics/

Reviewers' comments:

Reviewer's Responses to Questions

**Comments to the Author**

1. Is the manuscript technically sound, and do the data support the conclusions?

Reviewer #1: Yes

Reviewer #2: Partly

2. Has the statistical analysis been performed appropriately and rigorously? 

Reviewer #1: Yes

Reviewer #2: No

3. Have the authors made all data underlying the findings in their manuscript fully available?

Reviewer #1: Yes

Reviewer #2: No

4. Is the manuscript presented in an intelligible fashion and written in standard English?

Reviewer #1: No

Reviewer #2: Yes

5. Review Comments to the Author

Reviewer #1: The paper is concerned with the demonstration of the way a simulation of political discourse can utilize relationships between the variables affecting political engagement, and through those relations provide a correspondence between the variables of the social system and those of a thermodynamic system. The formulation proposed by the paper is significant concerning social physics to view social phenomena with physical principles.

Please kindly find below my recommendations to improve the paper:

The novelty of the paper can be stated more clearly.

The abstract can be revised to show the objective of the paper in a more evident way.

The distinctive parts of the paper can be stated so that the difference of this paper with previous ones can be shown.

I would also like to recommend English editing. For example, the first sentence in the abstract can be shortened or edited to make it clearer in meaning.

The authors can check if all the figures have been cited appropriately in the text.

Yours faithfully,

Reviewer #2: Review on manuscript:

„A Thermodynamic Definition and Equation of State for a Model of Political Election

Cycles”

--

The manuscript adresses new methodology for analysis of political elections processes. Political

election process relies on the metodology of classical statistical physics that is highly interlinked

with thermodynamics.

The concept of temperature is defined as dQ=dS/T so T=dS/dQ, where dQ stands for the transfered

heat or change of Hamitlonian energy. Such reasoning is not included in presented manuscript.

Therefore usage of term temperature is not fully justified.

The table with defined variables C(i,j,t)=C(x,y,t), M(i,j,t), I(i,j,t) shall be included for clarification

of manuscript. From the text we know that :

a) „A matrix is formed from the voting affiliation values, C, for each agent at their lattice positions

for each time point”,

b) „I is another matrix which holds the state of the ideological mismatch of an agent with their

locality”,

c) „ The matrix, M, contains the value for the voting action of each agent in the lattice at each time

point”.

Closer inspection of used quantities show certain similarity to Ising model and mean field theories.

Scalar field C is noting else equivalent to spin up or spin down in Ising model (with possible

accounting of fact that spin can have -3/2,-2/2,-1/2,0,1/2,2/2,3/2) etc.

The A shows effective local field (mean field) or its lack (as Superconducting Order Parameter or

Magnetization), while M accounts for possible gradient of me of I effective field.

In case of Ising model energy of a configuration σ is given by the Hamiltonian function

H(σ)=−∑〈 〉 i,j J(i,j)σiσj−μ∑j hjσj ,

where sums are over N by N lattice.

Therefore σi(x,y,t)is analogical to A(x,y,t) field, Step Function of (hi(x,y,t)) field is analogical to

M(i,j,t) field, while I(x,y,t) field is bit analogical to gradient of I(x,y,t) or momentum in quantum

mechanical terms. Coarse graing procedure is commonly known in statistical physics and can be

identified in Authors CMI model.

In such a way Hamiltonian for used „CMI” model (C(x,y,t), M(x,y,t), I(x,y,t)) shall be identified.

Then partition function denoted by Z shall be determined. Even Wikipedia gives broad description

of this metodology (https://en.wikipedia.org/wiki/Ising_model

https://farside.ph.utexas.edu/teaching/329/lectures/node110.html ,

https://en.wikipedia.org/wiki/Partition_function_(statistical_mechanics) ).

Equation of State is not fully justified. The manuscript cites:

T=exp(n_total), P=exp(n_pheripheral), V=exp(n_political), PV=T.

Therefore, manuscript implies operator equation: exp(n_pheripheral)exp(n_political)=exp(n_total).

How such operator equation can be justfied.

The Question is operator equation n_pheripheral+n_political=n_total.

If commutator [n_pheripheral,n_political] is equal to zero?

Schroedinger equation indeed is operator equation:

Ekinetic=[p^2/(2m)], Epot=V(x), Etotal=Ekinetic+Epot.

So if commutator [Ekinetic,Epot]=0 we can write

exp(Ekinetic)exp(Epot)=exp(Etotal) and nonzero commutator [Ekinetic,Epot] not equal 0, so

exp(Ekinetic+Epot)=exp(Etotal)

In Quantum Mechanics we have for example

H(t)|psi(x,t)>=i*hbar (d/dt)|psi(x,t)> leading to |psi(x,t)>=exp[(1/i*hbar)Int(H(t1),t0,t)]|psi(x,t0)>.

The title supposed to be changed into „A Possible Thermodynamic Definition and Equation of State

for a Model of Political Election Cycles”.

The manuscript shall cite additional positions:

P1) Van Kampen, „Stochastic Processes in Physics and Chemsistry”.

→ proper definition of temperature

P2) „Thermodynamics in Stochastic Conway’s Game of Life”

[ https://www.mdpi.com/2410-3896/8/2/47 ]→ methodology of thermodynamical description of

Game (as Election Process) is given with proper definition of temperature for given cellular

automata system.

P3) „Equivalence between finite state stochastic machine, non-dissipative and dissipative tight-

binding and Schrödinger model”(doi: 10.1016/j.matcom.2023.02.018 ) → methodology of mapping

Classical Statistical Physics Problem to Quantum Mechanics is Specified.

Major revision of article is required.

Temperature needs to be properly defined.

6. PLOS authors have the option to publish the peer review history of their article (what does this mean?). If published, this will include your full peer review and any attached files.

Reviewer #1: No

Reviewer #2: No

---

## [Author Response · Author response to Decision Letter 0]

24 Jan 2024

Response to Reviewer 1

(overall summary of changes): The authors are grateful for the comments and address each of the points.

• 1. "The novelty of the paper can be stated more clearly."

– (reply to the reviewer) This has been addressed within in the Introduction where more language was added to emphasize the novelty of the approach.

• 2. "The abstract can be revised to show the objective of the paper in a more evident way. The distinctive parts of the paper can be stated so that the difference of this paper with previous ones can be shown."

– (reply to the reviewer) The this has been addressed in the abstract and Introduction now.

• 3. "I would also like to recommend English editing. For example, the first sentence in the abstract can be shortened or edited to make it clearer in meaning." – (reply to the reviewer) This has been edited.

• 4. "The authors can check if all the figures have been cited appropriately in the text." – (reply to the reviewer) The figure references have now all been checked.

Response to Reviewer 2

(overall summary of changes): The authors are grateful for the comments and address each of the points.

• 1. "The concept of temperature is defined as dQ=dS/T so T=dS/dQ, where dQ stands for the transfered heat or change of Hamitlonian energy. Such reasoning is not included in presented manuscript. Therefore usage of term temperature is not fully justified.

– (reply to the reviewer) The overall aim is to produce a dimensionality reduction from the multiple variables of the microstates to more descriptive variables in a macroscopic description. The volume and pressure are defined the system. As such the dimensionality reduction can be achieved through an equation of state connecting the volume and the pressure and mandating a variable equivalent to temperature. Since the correlation between the volume and the pressure is categorically defined via the political vs the non-political actions the temperature definition must be such that it complies with this behaviors. While there is no singular way to define the temperature, in order to adapt the model to resemble an equation of state of an ideal gas the temperature would have to be analogous to the product of the volume and the pressure. Given the additive properties of the pressure and the volume on possible solution to transform this addition to a multiplication was to use an exponential operator. Therefore by doing the above we achieve that the microscopic behavior is adequately correlated to the pressure and the volume. And the pressure and the volume conform to a behavior similar to an equation of state of an ideal gas. As such the temperature is defined. While we do understand that this is not the only possibly way to define a thermodynamic system upon a social system we believe that it is adequate as it is able to capture the phenomena produced in the simulation as well as resemble standard thermodynamic equations. In this formulation of the model the energy is not defined and as a consequence the correlation of the temperature and

1

the energy is not a necessary. Considering the energy is part of future work and now mentioned in the final words of the text.

• 2. "In such a way Hamiltonian for used „CMI” model (C(x,y,t), M(x,y,t), I(x,y,t)) shall be identified. Then partition function denoted by Z shall be determined. Even Wikipedia gives broad description of this metodology. Equation of State is not fully justified. The manuscript cites: T=exp(n_total), P=exp(n_pheripheral), V=exp(n_political), PV=T. Therefore, manuscript implies operator equation: exp(n_pheripheral)exp(n_political)=exp(n_total). How such operator equation can be justfied. The Question is operator equation n_pheripheral+n_political=n_total.

If commutator [n_pheripheral,n_political] is equal to zero?"

– (reply to the reviewer) Although a very interesting idea to formulate the Hamiltonian, the scope is directed at only the thermodynamic perspective. The Hamiltonian could potentially allow more accurate representation of social dynamics among agents but for this work the thermodynamics is the main scope. As such the Z can be omitted without considering possible further representations. The equation of state is justified using the above response which is based on the aspect that the energy for this model is not defined. It is a possible avenue for future work to consider the energy formulation based on agent state but in this approach no such proposals are made. This allows the current equation of state to be valid given the variables included. The relationship n_pheripheral+n_political=n_total is contained in the model and both quantities can be known allowing for those statements to hold.

• 3. "Schroedinger equation indeed is operator equation: Ekinetic = [p2/(2m)],Epot = V (x),Etotal = Ekinetic+Epot. So if commutator [Ekinetic,Epot]=0 we can write exp(Ekinetic)exp(Epot)=exp(Etotal) and nonzero commutator [Ekinetic,Epot] not equal 0, so exp(Ekinetic+Epot)=exp(Etotal) In Quantum Mechanics we have for example H(t)|psi(x,t)>=i*hbar (d/dt)|psi(x,t)> leading to |psi(x,t)>=exp[(1/i*hbar)Int(H(t1),t0,t)]|psi(x,t0)>. The title supposed to be changed into „A Possible Thermodynamic Definition and Equation of State for a Model of Political Election Cycles”."

– (reply to the reviewer) In QM the kinetic and potential energy are independent in their definitions and as such their commutator is zero. Similarly in our case, the pressure and the volume can, by definition, be know simultaneously. Therefore, their commutator is zero and therefore the equation exp(P)exp(V) = exp(T) can be written. This also conforms to the simulation rules as well.

• 4. "The manuscript shall cite additional positions: P1) Van Kampen, „Stochastic Processes in Physics and Chemsistry”. → proper definition of temperature

P2) „Thermodynamics in Stochastic Conway’s Game of Life” [ https://www.mdpi.com/24103896/8/2/47 ]→ methodology of thermodynamical description of Game (as Election Process) is given with proper definition of temperature for given cellular automata system.

P3) „Equivalence between finite state stochastic machine, non-dissipative and dissipative tightbinding and Schrödinger model”(doi: 10.1016/j.matcom.2023.02.018 ) → methodology of mapping"

– (reply to the reviewer) These highly relevant papers have now been included in the text emphasizing their relevance and merit.

---

## [Decision Letter · Decision Letter 1]

20 Feb 2024

PONE-D-23-39285R1A Possible Thermodynamic Definition and Equation of State for a Model of Political Election CyclesPLOS ONE

Dear Dr. Mantzaris,

Thank you for submitting your manuscript to PLOS ONE. After careful consideration, we feel that it has merit but does not fully meet PLOS ONE’s publication criteria as it currently stands. Therefore, we invite you to submit a revised version of the manuscript that addresses the points raised during the review process. Please submit your revised manuscript by Apr 05 2024 11:59PM. If you will need more time than this to complete your revisions, please reply to this message or contact the journal office at plosone@plos.org. Please include the following items when submitting your revised manuscript:A rebuttal letter that responds to each point raised by the academic editor and reviewer(s). You should upload this letter as a separate file labeled 'Response to Reviewers'.A marked-up copy of your manuscript that highlights changes made to the original version. You should upload this as a separate file labeled 'Revised Manuscript with Track Changes'.An unmarked version of your revised paper without tracked changes. You should upload this as a separate file labeled 'Manuscript'.If applicable, we recommend that you deposit your laboratory protocols in protocols.io to enhance the reproducibility of your results. Protocols.io assigns your protocol its own identifier (DOI) so that it can be cited independently in the future. For instructions see: https://journals.plos.org/plosone/s/submission-guidelines#loc-laboratory-protocols. Additionally, PLOS ONE offers an option for publishing peer-reviewed Lab Protocol articles, which describe protocols hosted on protocols.io. Read more information on sharing protocols at https://plos.org/protocols?utm_medium=editorial-email&utm_source=authorletters&utm_campaign=protocols.

We look forward to receiving your revised manuscript.

Kind regards,

Mohammad Tariq

Academic Editor

PLOS ONE

Journal Requirements:

**Additional Editor Comments:**

**      The manuscript is in good shape now. The authors need to address the comments of the reviewer #2.**

Reviewers' comments:

Reviewer's Responses to Questions

**Comments to the Author**

1. If the authors have adequately addressed your comments raised in a previous round of review and you feel that this manuscript is now acceptable for publication, you may indicate that here to bypass the “Comments to the Author” section, enter your conflict of interest statement in the “Confidential to Editor” section, and submit your "Accept" recommendation.

Reviewer #1: All comments have been addressed

Reviewer #2: All comments have been addressed

2. Is the manuscript technically sound, and do the data support the conclusions?

Reviewer #1: Yes

Reviewer #2: Yes

3. Has the statistical analysis been performed appropriately and rigorously? 

Reviewer #1: Yes

Reviewer #2: I Don't Know

4. Have the authors made all data underlying the findings in their manuscript fully available?

Reviewer #1: Yes

Reviewer #2: Yes

5. Is the manuscript presented in an intelligible fashion and written in standard English?

Reviewer #1: Yes

Reviewer #2: Yes

6. Review Comments to the Author

Reviewer #1: The structure of the paper is well-organized. The paper is well-written with steps explained in good details. The novel aspects have been very duly explained as well. The recommendations have been integrated into the study which has been improved accordingly.

The paper can be considered suitable for publication.

Yours faithfully,

Reviewer #2: The improvements of manuscript has been conducted, but not fully in all suggested aspects.

The most troublesome is definition of temperature and thermodynamic cycle assigned to Model of Political

Election Cycles. Apriori Authors suggest the simplest equation of state pV=nRT as it is the case of ideal gas (as effectively describing political election cycle). Most gases or physical systems have much more complicated equation of state. Furthermore relation dS=dE/T or equivalently dE/dS=T is the best phenomenological definition of temperature T. Such definition is not undertaken in manuscript since Authors have not proposed any definition of Hamiltonian or energy in system of interacting agents in election cylce. One can report that definition of entropy was given in manuscript.

Due to this fact I recommend to add one sentence in conclusion section:

"Further justification of usage of thermodynamic cycles methodology in political election cycle shall be attempted with

introduction of definition of effective Hamiltonian (that is dependent on (C,M,I) ), so Hamiltonian change dE accompanied with entropy change dS will lead to temperature dependence expressed by formula T=dE/dS " .

Once this sentence in Conclusion section is added I recommend article for publication.

7. PLOS authors have the option to publish the peer review history of their article (what does this mean?). If published, this will include your full peer review and any attached files.

Reviewer #1: **Yes: **Yeliz Karaca

Reviewer #2: No

---

## [Author Response · Author response to Decision Letter 1]

20 Feb 2024

The text requested by Reviewer 2 is now included in the Conclusion Section as requested.

---

## [Editor Report · Decision Letter 2]

23 Feb 2024

A Possible Thermodynamic Definition and Equation of State for a Model of Political Election Cycles

PONE-D-23-39285R2

Dear Dr. Mantzaris,

We’re pleased to inform you that your manuscript has been judged scientifically suitable for publication and will be formally accepted for publication once it meets all outstanding technical requirements.

Kind regards,

Mohammad Tariq

Academic Editor

PLOS ONE
---

## [Editor Report · Acceptance letter]

27 Feb 2024

PONE-D-23-39285R2 

PLOS ONE

Dear Dr. Mantzaris, 

I'm pleased to inform you that your manuscript has been deemed suitable for publication in PLOS ONE. Congratulations! Your manuscript is now being handed over to our production team.

Kind regards, 

on behalf of

Dr. Mohammad Tariq 

Academic Editor

PLOS ONE